# Clinical Aspects of Genetic and Non-Genetic Cardiovascular Risk Factors in Familial Hypercholesterolemia

**DOI:** 10.3390/genes13071158

**Published:** 2022-06-27

**Authors:** Eszter Berta, Noémi Zsíros, Miklós Bodor, István Balogh, Hajnalka Lőrincz, György Paragh, Mariann Harangi

**Affiliations:** 1Division of Metabolism, Department of Internal Medicine, Faculty of Medicine, University of Debrecen, H-4032 Debrecen, Hungary; eberta@belklinika.com (E.B.); zsiros.noemi@belklinika.com (N.Z.); lorincz_hajnalka@belklinika.com (H.L.); paragh@belklinika.com (G.P.); 2Division of Endocrinology, Department of Internal Medicine, Faculty of Medicine, University of Debrecen, H-4032 Debrecen, Hungary; bodor@belklinika.com; 3Division of Clinical Genetics, Department of Laboratory Medicine, Faculty of Medicine, University of Debrecen, H-4032 Debrecen, Hungary; balogh@med.unideb.hu

**Keywords:** familial hypercholesterolemia, genetic factors, risk stratification, endocrine diseases, high-density lipoprotein, thyroid, diabetes mellitus, polycystic ovary syndrome, growth hormone

## Abstract

Familial hypercholesterolemia (FH) is the most common monogenic metabolic disorder characterized by considerably elevated low-density lipoprotein cholesterol (LDL-C) levels leading to enhanced atherogenesis, early cardiovascular disease (CVD), and premature death. However, the wide phenotypic heterogeneity in FH makes the cardiovascular risk prediction challenging in clinical practice to determine optimal therapeutic strategy. Beyond the lifetime LDL-C vascular accumulation, other genetic and non-genetic risk factors might exacerbate CVD development. Besides the most frequent variants of three genes (*LDL-R*, *APOB*, and *PCSK9*) in some proband variants of other genes implicated in lipid metabolism and atherogenesis are responsible for FH phenotype. Furthermore, non-genetic factors, including traditional cardiovascular risk factors, metabolic and endocrine disorders might also worsen risk profile. Although some were extensively studied previously, others, such as common endocrine disorders including thyroid disorders or polycystic ovary syndrome are not widely evaluated in FH. In this review, we summarize the most important genetic and non-genetic factors that might affect the risk prediction and therapeutic strategy in FH through the eyes of clinicians focusing on disorders that might not be in the center of FH research. The review highlights the complexity of FH care and the need of an interdisciplinary attitude to find the best therapeutic approach in FH patients.

## 1. Introduction

Familial hypercholesterolemia (FH) is the most common monogenic disorder, affecting millions of people worldwide. Based on our previous studies the estimated prevalence of FH in European countries is 1:340 [1]. FH is caused by inherited autosomal-dominant defects of the low-density lipoprotein (LDL) metabolism. There are three important genetic loci linked to FH, with the vast majority (approximately 88%) of cases being caused by mutations in the *LDL receptor* (LDLR) gene. Based on its high prevalence, patients with FH are seen frequently in the everyday clinical practice by general practitioners, cardiologists, pediatrists, lipidologists, and endocrinologists. Due to the effective lipid-lowering medications, the survival of FH patients is significantly better today than some decades ago, but this also means that these elderly FH patients might have other concomitant chronic diseases leading to altered clinical manifestations and unusual symptom patterns. These alterations in the clinical appearance of FH may lead to diagnostic error, incorrect cardiovascular risk evaluation and, thus, therapeutic misconception. Therefore, the better knowledge of the broad spectrum of parameters that can modify the disease course in FH might be crucial in the optimization of the individual therapeutic strategy.

## 2. Familial Hypercholesterolemia as Cardiovascular Risk Factor

The most important characteristic of FH is accelerated atherosclerosis due to the long-term, substantial exposure to high concentrations of circulating LDL. FH patients without treatment are at approximately 10–20-fold increased risk for atherosclerotic complications such as coronary artery disease (CAD). This hazard can be significantly reduced with early diagnosis and treatment of FH but remains still higher compared to the non-FH population. However, despite the high cumulative LDL-C burden, not all FH patients will develop cardiovascular disease (CVD) to the same extent, resulting in wide phenotypic heterogeneity [2]. It should be noted that standard assessment tools (e.g., Framingham) do not accurately quantify the risk in these patients.

Indeed, the characteristics and prevalence of cardiovascular events can remarkably vary between patients with FH, even among those who have the same genetic mutation [3]. By now, it has become clear that the clinical manifestations of FH and its cardiovascular outcomes are most likely caused by several factors, including genetic and non-genetic (environmental and metabolic) ones [4].

## 3. Genetic Causes of Familial Hypercholesterolemia

FH is traditionally classified into a “heterozygous” and a “homozygous” clinical phenotype. The heterozygous phenotype is inherited in an autosomal dominant manner and has an LDL-C concentration between 5 and 12 mmol/L (193–464 mg/dL), tendon xanthomata can be observed, and CAD before 55 years of age is common. The most important cause of the heterozygous phenotype is a mutation in the *LDLR*. Sometimes a mutation disrupting the binding of apolipoprotein B100 (apoB100) to the *LDLR* is encountered. Some other proteins, such as neural apoptosis regulated convertase 1 (*NARC1*), the product of the gene proprotein convertase subtilisin/kexin type 9 (*PCSK9*) has been identified [5].

Until now, over 2100 *LDLR* and 250 *APOB* mutations have been described in the literature (Professional Human Gene Mutation Database, 2021.3 release). Interestingly, no mutational hotspots have been identified in the *LDLR* gene; the genetic variants are widely spread from the promoter region to the very last exon of the gene. However, pathogenic variants occur more frequently in some regions, such as the ligand-binding domain and the epidermal growth factor-like domain [6].

Individuals with a clinical diagnosis of definite FH are found to present causal mutations in only 60–80% of cases. This observation may be explained by mutations in established genes that have yet to be identified, novel genes, or presence of an FH phenocopy [7]. 

The most relevant information about the genetics of familial hypercholesterolemia are summarized excellently in a former review published by Vrablik et al. [8]. 

## 4. Concomitant Genetic Abnormalities 

Beside mutations in *LDLR*, *APOB*, and *PCSK9* genes, FH variable phenotypes can depend on several modifier factors that also include variants in several lipid-related genes, as well as variants causing different genetic dyslipidemias [9].

## 5. Genes Implicated in Lipoprotein Metabolism

In a previous study, Guay et al. analyzed the epigenetic profile of four additional genes participating in lipoprotein metabolism: *ATP-binding cassette transporter G1 (ABCG1;* 21q22.3), *hepatic lipase (LIPC*; 15q21.3), *phospholipid transfer protein (PLTP*; 20q13.12), and *scavenger receptor B1 (SCARB1*; 12q24.31). Sex-specific multivariable linear regression models showed that new and previously reported epipolymorphisms of the studied genes (*ABCG1-CpGC3*, *LIPC-CpGA2*, *mean PLTP-CpGC*, *LPL-CpGA3*, *CETP-CpGA2*, *and CETP-CpGB2*) significantly contributed to variations in plasma lipid levels independently of traditional predictors such as age, waist circumference, blood pressure, fasting plasma lipids, and glucose levels. The authors concluded that epigenetic perturbations of key lipoprotein metabolism genes are associated with the plasma lipid levels, contribute to the inter-individual variability, and might at least partially explain the missing heritability of the altered plasma lipid levels in FH [10]. Presumably, a wide application of next generation sequencing using extended gene panels will reveal the real frequencies of FH patients carrying variants in other genes and might explain the different phenotypes [9].

## 6. Concomitant Diseases

Simultaneous presence of concomitant diseases is associated with adverse health outcomes, more complex clinical management, and additional health care costs in FH [11]. Some diseases, such as obesity, hypertension, and diabetes commonly overlap, but some of them were found to be independent of the other disorders. Although the effect of these concomitant diseases on the cardiovascular risk and outcomes is obviously unfavorable, their impact on the estimated CVD risk has not yet been completely clarified. Moreover, recommendations for the systematic screening of these disorders needs to be standardized. Indeed, some endocrine abnormalities are not in the limelight of the cardiologists, although these hormonal disorders might also contribute to the enhanced atherogenesis in FH patients (Table 1). 

## 7. Coexisting Dyslipidemias with FH

### 7.1. Role of High-Density Cholesterol Levels

The total to high-density lipoprotein (HDL) total cholesterol (TC) ratio (TC/HDL-C) as a CVD risk factor was focused on recently [12]. Instead of an LDL-C level of 130 mg/dL as the cut-off point, using a TC/HDL-C ratio of 5 was associated with superior specificity and accuracy in predicting future coronary heart disease (CHD) [13]. A defective HDL-driven cholesterol efflux may also be associated with low levels of HDL cholesterol in homozygous FH (HoFH). However, ambivalent results regarding HDL-C concentrations have been reported in heterozygous FH (HeFH) populations [14]. A former study proved that multiple, modestly penetrant but highly prevalent polymorphisms might explain a substantial part of the variation in HDL-C plasma levels in a cohort of HeFH patients. Overall, the effects of polymorphisms involved in HDL metabolism can explain 12.5% of the variation in the plasma HDL-C. Interestingly, when sex and several environmental factors were considered, an impressive 32.5% of variation in HDL-C levels was found [15]. Anyhow, low HDL-C concentration has been shown to be an independent risk factor for the development of CVD [14]. Low HDL-C phenotype has been associated with a 37% relative increase in the CVD risk and was suggested to be a strong marker of preclinical carotid atherosclerosis in HeFH patients [16]. Moreover, HDL functions are often harmed in FH [17] due to additive changes in proteins, enzymes, lipids, and microRNA which lead to the loss of their atheroprotective properties [18].

### 7.2. Lipoprotein (a)

Similar to LDL particles, lipoprotein (a) (Lp(a)) is a lipoprotein that is produced by the covalent bonding of apolipoprotein(a) (apo(a)) to ApoB. Former studies have demonstrated that high plasma Lp(a) concentrations are associated with an elevated risk of arteriosclerotic cardiovascular disease, most probably due to the pro-atherogenic and pro-inflammatory effects of Lp(a) [19]. Therefore, the 2019 European guidelines on dyslipidemia recommend measuring Lp(a) at least once in a lifetime in both non-FH and FH patient populations [20]. A former meta-analysis has shown that a high Lp(a) value plays an important role in the development of CVD in FH individuals [21]. Several previous studies reported that serum Lp(a) levels were significantly higher in FH than in non-FH subjects, regardless of their CVD history [22,23,24]; however, other reports failed to corroborate these findings [25,26]. Indeed, it has not been confirmed that *LDLR* or other mutations resulting in heterozygous FH phenotype affect Lp(a) levels. At the same time, Lp(a) values in some FH individuals can depend on other genetic factors including variations in genes encoding plasminogen, lipoprotein, toll-like and scavenger receptors, and lectins. Since a portion of Lp(a) is catabolized by LDLR, dysfunctional LDLR due to the concomitant genetic variants leads to an increase in Lp(a) [27].

Several lipid-lowering agents (mostly aimed at lowering LDL-C) have proven useful in reducing Lp(a) levels. ApoB secretion inhibitor mipomersen and microsomal transfer protein inhibitor lomitapide are indicated exclusively in HoFH, but the PCSK9 inhibitor monoclonal antibodies (alirocumab and evolocumab) and the small interfering RNA inclisiran are commonly used in HeFH, and are also able to decrease the Lp(a) level by 2025% [28]. Selective lipoprotein apheresis, although not widely available, has also proven very effective in lowering the Lp(a) level [29,30]. Furthermore, the use of specifically targeted therapies including antisense oligonucleotides APO(a)Lrx were able to reduce Lp(a) from 35% to over 80% with generally modest injection site reactions [31]. 

### 7.3. The Role of Hypertriglyceridemias

Former epidemiological studies showed that the elevated fasting plasma triglyceride (TG) concentration also contributes to the development of atherosclerosis and is an independent risk factor for CHD [32,33]. Although one of the possible explanations of heterogeneity in the manifestation of atherogenic disease in FH patients could lie in the elevated TG levels, the significance of TG as a risk factor is significantly underestimated in FH. 

In fasting state, TGs are not usually elevated in FH, suggesting that the production and clearance of chylomicrons (CMs) are normal. However, in the absence of LDLR, the upregulation of very low-density lioporotein receptor (VLDLR) in the liver has been observed, resulting in an increased secretion of VLDL [34]. In consequence, significant, 50–109% increases have been reported in VLDL apo B production in heterozygous and homozygous FH, respectively [35]. Since TG-rich lipoproteins, including CMs are partially catabolized by the hepatic LDLR in postprandial state, the reduced activity of LDLR leads to increased accumulation of CM remnants [36]. Patients with HeFH present elevated plasma concentrations of postprandial TG-rich lipoprotein remnants, including those of intestinal origin, which might be causally related to atherosclerosis [37].

Moreover, cases with coexisting dysbetalipoproteinemia caused by ApoE2 homozygosity and defective LDL receptor heterozygosity result in severe type III hyperlipidemia [38]. Fortunately, this type of coexistence has been estimated to be just around 2:100,000; therefore, representing an unusual cause of type III hyperlipidemia. Coexistence of the highly polygenic burden of single nucleotide polymorphisms that raise TG levels by production or clearance mechanism and FH can also lead to severe hypertriglyceridemia and atypical presentation of HeFH [39]. 

It must be highlighted that secondary causes of hypertriglyceridemia including high-fat diet, overweight and obesity, diabetes mellitus, alcohol consumption, renal diseases, especially nephrotic syndrome and hepatic disorders may also be associated with FH. Administration of medications such as estrogen, isotretinoin, immunosuppressant therapy, thiazides, β blockers, atypical antipsychotics, and glucocorticoids, administered in FH patients for concomitant diseases, can also influence the TG metabolism. Less frequent causes of hypertriglyceridemia such as endocrine diseases including Cushing’s syndrome, acromegaly, hypothyroidism, lipodystrophies, and autoimmune diseases can also be present in patients with FH [40]. 

## 8. Hypertension

Hypertension is a known independent and important risk factor for CVD in genetically confirmed FH patients [41], but data on the prevalence of hypertension in FH patient populations is extremely variable. In Polish patients, high prevalence of arterial hypertension was reported in definite (69.4%), probable (70.7%), and possible (72.6%) FH identified using the Dutch Lipid Clinic Network Score (DLCNS) [42]. In line with some other Eastern European countries, the prevalence of hypertension was extremely high (86.3%) in a Hungarian FH population [24]. In total, 50.8% and 59.2% of hypertension prevalence had been described in FH patients from Romania and Karelia, respectively [43] [44]. On the other hand, an Iranian FH registry showed similar prevalence of hypertension in possible FH patients (25.3%) and in the general population (25.2%) [45]. In an Italian study, 16.2% and 23.8% FH prevalence was detected in males and females, respectively [46]. Interestingly, a relatively low, 17% hypertension prevalence was identified in a Mexican FH cohort [47], in another previous study hypertension was found in only 11% of FH patients [48]. 

Interestingly, based on a molecular polymorphism analysis of genes involved in the renin–angiotensin system, it was found that the diastolic blood pressure was independently related to CVD risk in FH patients. The angiotensin-II type I receptor A1166C polymorphism may interact with severe hypercholesterolemia and other risk factors to increase the risk of CHD in FH patients [49]. 

## 9. Smoking

The influence of smoking habits on atherosclerosis such as inflammation, fat peroxidation, brown fat metabolism, and coronary artery calcification has been thoroughly evaluated in the general population. In addition, smoking interacts with many gene variants by modulating the risk of CAD, CVD, and stroke [50]. Various studies have proved that smoking may influence apolipoprotein concentrations and lipoprotein particle sizes, and a gene *x* smoking interaction might influence the lipid–lipoprotein metabolism. A recent large, multi-ancestry, genome-wide gene–smoking interaction study has reported 13 novel loci associated with lipid levels; however, interpreting the results is challenging in the case of gene–environment interaction studies as the environment factors are difficult in order to control adequately the studied disorders, moreover, lipid–lipoprotein phenotypes are heterogeneous [51].

In a novel study, PPARα-L162V, heterozygous loss-of function lipoprotein lipase (*LPL*) mutation, Apo E4 allele, or Apo E2/2 genotype were proved to be associated with a plasma apoB concentration increase, by their action on specific apoB-containing lipoproteins. Furthermore, reduced LPL activity and increased hepatic lipase activity has been observed among smokers [52,53].

## 10. Endocrine Disorders

### 10.1. The Correlation between Thyroid Hormones and Lipid Homeostasis 

Thyroid hormones, 3,3′,5,5′-tetraiodo-L-thyronine (T4) and 3,5,3′-triiodo-L-thyronine (T3) affect robustly the basic energy expenditure and lipid metabolism by stimulating the mobilization and degradation of lipids and de novo fatty acid synthesis of the liver [54,55]. Thyroid hormones increase cholesterol synthesis via the rate-limiting enzyme 3-hydroxy-3-methylglutaryl coenzyme A reductase (HMG-CoAR) and stimulate its removal by enhancing the expression of LDLRs in the liver [56]. The synthesis of lipids is regulated by sterol regulatory element-binding proteins (SREBPs) and carbohydrate response element binding protein (ChREBP), which are considered the most important lipogenic transcription factors [55]. SREBPs positively regulate the expression of LDL receptor and cholesterol synthesis; thus, thyroid hormones have a modulating effect on the hepatic lipogenesis through *SREBP-1* and *ChREBP* gene expression alterations [54]. Thyroid hormones also increase the activity of hepatic lipase, lipoprotein lipase, and cholesterol ester transfer protein [54] (Figure 1).

### 10.2. Hypothyroidism

Hypothyroidism is a common disease with a prevalence of 1.4–13% among patients with hyperlipidemia [54,58]. Overt and subclinical hypothyroidism are common causes of secondary hyperlipidemia, as lipid levels show a gradual elevation in accordance with elevated thyroid stimulating hormone (TSH) levels. The dyslipidemia in the hypothyroid state is the consequence of increased synthesis to degradation rate, with elevated levels of total cholesterol, mainly LDL-C. High serum concentrations of TGs, intermediate-density lipoproteins, ApoA, and ApoB are also detected [59,60]. Hyperlipidemia and lipid peroxidation, contributing to oxidative stress and the decreased rate of synthesis and catabolism of fatty acids together with the blunted lipolytic sensitivity of white fat cells in hypothyroidism, may lead to the development of CHD [54,61]. In clinical practice, a period of 4–6 weeks of thyroxin replacement therapy is usually needed to correct dyslipidemia in overt hypothyroidism, changes in serum lipoproteins in hypothyroidism are correlated with changes in the free T4 [62]. Furthermore, in hypothyroidism hepatic triglyceride lipase activity decreases, which leads to the elevation of TG serum levels [56]. In addition, the prevalence of metabolic syndrome, which represents considerable cardiovascular risk, was reported to be 44% in patients with hypothyroidism and 35% among patients with subclinical hypothyroidism compared with 33% in the control group in the study of Erdogan et al. [63]. Besides alteration of lipid homeostasis, both hypo- and hyperthyroidism can slightly increase the risk of hypertension [64,65,66]. Arterial stiffness is increased in hyperthyroidism due to genomic and non-genomic action of thyroxin targeting membrane ion channels and endothelial nitric oxide synthesis of vascular smooth muscle and endothelial cells [67,68]. Overt hypothyroidism (elevated TSH with free thyroid hormones in the normal range) leads to elevated diastolic blood pressure in approximately 30% of the patients [64]. The effect of hypothyroidism on accelerated atherosclerosis is well known and investigated, but there is a lack in therapeutic guidelines regarding subclinical hypothyroidism treatment among patients affected by hyperlipidemias and diabetes mellitus, the latter having a bi-directional relationship with thyroid disorders [54,56]. Furthermore, a substantial number of studies indicated a beneficial response in euthyroid patients with TSH between 2.5 and 4.5 concerning atherogenic lipid parameters, impaired endothelial function, and intima media thickness [69]. In a frail population such as FH patients regarding CVD, our knowledge on the prevalence of thyroid disorders and therapeutic consequences seems incomplete.

### 10.3. Diabetes Mellitus and Familial Hypercholesterolemia

In the FH population, the prevalence and pathomechanism of diabetes development is not known in detail, neither is its impact on CVD risk. 

Diabetes is a well-established risk factor for CVD, which leads to a 2–4-fold higher CVD risk in adult diabetic individuals compared to non-diabetics in the general population [70]. The risk of CVD rises with worsening glycemic control in an independent manner from other traditional risk factors. A global increase in sedimentary lifestyle, the aging of the population, and the escalating rate of obesity among adults and children are the major cause of the rapid increase in the diabetes prevalence worldwide [71]. Diabetes is associated with a 75% increase in mortality, mainly due to its vascular complications, a large part of excess mortality is caused by CVD [72]. Beside premature death, diabetes-related micro- and macrovascular complications including CHD, cardiomyopathy, cerebrovascular disease, peripheral vascular disease, chronic kidney failure, diabetic retinopathy, polyneuropathy, and cardiovascular autonomic neuropathy worsen the quality of life and can also lead to disability [71,72]. The effects of more stringent diabetes control on CVD morbidity and mortality had led to more focused current diabetes guidelines. 

### 10.4. Diabetic Dyslipidemia

In type 2 diabetes mellitus (T2DM), macrovascular complications develop through different pathogenic pathways that include hyperglycemia, dyslipidemia, and insulin resistance leading to impaired platelet function, endothelial, vascular smooth muscle cell dysfunction, and abnormal coagulation [72]. 

The development of vascular complications may be present early, in pre-diabetes stages. Fasting plasma glucose (FPG) is associated with elevated CVD risk even below the threshold for diabetes (7 mmol/L or 126 mg/dL) [73]. Diabetic dyslipidemia is characterized by high TGs and low HDL-C and can precede the development of hyperglycemia by years. 

In T2DM, insulin resistance activates the intracellular hormone-sensitive lipase and consequently the release of non-esterified fatty acids from TGs stored in visceral adipose tissue becomes increased together with the hepatic TG production [74]. The increased hepatic TG synthesis is associated with increased secretion of ApoB, while the normal inhibitory effect of insulin on hepatic ApoB production and TG secretion in VLDL is absent, resulting in larger and more TG-rich VLDL [75,76,77,78]. Furthermore, VLDL catabolism becomes reduced and lipoprotein lipase in the vascular endothelium may be downregulated in insulin resistant or deficient states which contributes to postprandial lipemia [79,80]

The angiopathic risk can be predicted with the ratio of HDL-C/ApoA1 [81]; moreover, HDL from T2DM patients stimulates tumor necrosis factor-α secretion from mononuclear cells from peripheral blood in vitro [82], the ability of which was more pronounced among samples from patients with CAD. A retrospective study of Russo et al. found low HDL-C and high TG levels to be independent risk factors for the development of diabetic kidney disease [83]. In T2DM, poor glycemic control results in functional and compositional alterations of small dense HDL. While lifestyle intervention improves the levels of HDL and large LDL particles [83,84,85,86]. 

### 10.5. Prevalence and Clinical Relevance of Diabetes in Familial Dyslipidemia

In an early study, 13 FH patients free from other cardiovascular risk factors (hypertension, overweight, glucose intolerance, high serum TG concentrations) had been compared to a matched group of healthy normocholesterolemic controls to assess their insulin action on carbohydrate and energy expenditure via a 2 h euglycemic hyperinsulinemic clamp. Among them, the elevated plasma LDL cholesterol levels did not affect carbohydrate metabolism, total insulin-mediated glucose disposal, carbohydrate and lipid oxidation rates [87]. 

Statins are associated with an increased risk for T2DM as HMG-CoAR increases the expression of LDL receptors in many tissues causing a rise in the cellular cholesterol uptake which leads to β-cell dysfunction. Statins increase the risk of T2DM in the general population by 9% in a dose-dependent manner [88]. Furthermore, a genetic variation at the *HMGCR* [NCBI Entrez Gene 3156] locus is also associated with T2DM. The transmembrane cholesterol uptake might be linked to the development of T2DM, and the impaired cholesterol uptake in FH might have a causative effect on the lower prevalence of T2DM among FH individuals [89]. Although FH patients are treated with high dose statins, the diabetogenic effect of statins is not remarkable among them [90]. In a study conducted on patients receiving long-term, decade-long statin therapy because of HeFH or familial combined hyperlipidemia (FCH), the development of diabetes was 14% in FCH and 1% in FH patients. In FH patients, neither statin treatment intensity nor type of statins used seemed to play a role in the development of T2DM [91].

As the CVD risk of mutation-positive FH patients with elevated LDL-C levels is 22-fold higher than of the non-FH individuals [92], the vascular compounds causing complications in diabetes might further worsen the patients’ survival. However, FH patients benefit from a strict clinical follow-up and lifestyle counseling due to their known elevated CVD risk. 

In a study conducted on 1412 patients from the FH Canada Registry, a total of 73 diabetic patients (5%) were found. The prevalence of CVD was higher among diabetic FH patients (45%) compared to nondiabetics (22%). The Montreal FH-SCORE which uses clinical predictors as age, gender, HDL-C, hypertension, and smoking was higher in the diabetic group than in the non-diabetic population. After the adjustment for the Montreal FH-SCORE diabetes was no longer a significant predictor of CVD suggesting that FH diabetic subjects may have many concomitant cardiometabolic risk factors [70].

In a cross-sectional analysis by Besseling et al. conducted in the Netherlands, the prevalence of type 2 diabetes was 1.75% among 25,000 FH patients compared to 2.93% in 38,000 unaffected relatives. Furthermore, diabetes was an independent CVD predictor among FH subjects in a multivariate analysis [89]. The prevalence of T2DM was lower in carriers of ApoB mutations, which results in a less severe phenotype compared to unaffected relatives. This difference was more pronounced in patients with *LDLR* mutations and the severity of the *LDLR* mutation was inversely associated with the prevalence of T2DM [89]. The overall prevalence of T2DM among the Old Order Amish was 4.6% in a study, and contrary to the results of Besseling et al., the R3527Q T allele was associated with an increased, not decreased, risk of T2DM, suggesting that ancestry and other genetic and epigenetic factors can affect FH phenotypes [93].

The prevalence of T2DM among FH patients of the Dyslipidemia Registry of the Spanish Arteriosclerosis Society was 5.9%, which was lower than the 9.4% prevalence in the age and sex-adjusted Spanish population. FH patients with diabetes were older and treated with statins for a longer time, had a higher body mass index (BMI), waist circumference, TGs, glycosylated hemoglobin, cardiovascular disease, and hypertension prevalence. Gene mutations and LDL cholesterol concentrations were not associated with T2DM prevalence in the Spanish study [90]. The protective role of FH, which leads to 40% lesser T2DM prevalence among FH patients was not affected by age and gender and reached statistical significance only in older groups. The lower risk of T2DM appeared in a large LDL-C range. However, in the Spanish study, LDL-C concentration and gene mutations did not affect T2DM risk; the presence or absence of causative gene mutations responsible for intracellular cholesterol uptake had a beneficial impact on diabetes risk if investigated in more numerous cohorts [89,90]. 

The results from the TERCET registry with 19,781 individuals considered having very-high CV risk and admitted with acute coronary syndrome showed lower diabetes prevalence among definite FH, probable FH, and possible FH patients (who were evaluated with DLCNS) compared to patients without FH (22.4%, 25.3%, 29.9% vs. 34.5%, respectively) [42]. 

In our recently published study using data mining methods, a total of 1,342,124 patients’ DLCNS were calculated, as a result 6 definite, 225 probable, and 11,706 possible FH patients were identified. The prevalence of diabetes among them was higher than the one found in the general population, 23.2, 21.8, and 21.9%, respectively, while diabetes prevalence in Hungary had been reported as 8135 cases/100,000 persons in 2011 [94]. Obesity was present in 8.9% of definite, 5.6% of probable, and 7.1% of possible FH patients. Uniquely, in our study the prevalence of hypothyroidism was also evaluated, being 8.9% among definite, 10.5% of probable, and 9.4% of possible FH patients [1]. In our study analyzing 590,500 Hungarian patients’ medical records, 459 FH patients were identified, 221 of them had data also available on Lp(a). Diabetes prevalence was 17.4%, higher than the 5.16% diabetes prevalence of non-FH patients. In the latter study, obesity prevalence was found to be 42%, more frequent among FH patients, than in the earlier work [24]. The prevalence of diabetes in our FH cohort was comparable to the prevalence reported in Polish, Romanian (13.1%), and Mexican (11.3%) FH patients [42]. Reported prevalence of diabetes was high among Turkish patients as well, 22.4% of patients in the A-HIT 2 Turkish FH registry had diabetes [29]. Among Slovakian patients, the prevalence of diabetes was 10.5% reported by Vohnout et al. [95]. 

The diabetes prevalence reported from Poland, Romania, Hungary, and Slovakia is considerably higher than the early data reported in Canada, the Netherlands, and Spain. The difference might be caused by the age of the studied cohort, socioeconomic status, and genetic influencing factors, as well as the lack of the previous knowledge of FH among patients and medical staff, as many of the studies involved newly diagnosed FH patients in the analysis [24]. 

In a novel study analyzing a database of diabetes centers across Bulgaria, DLCNS was calculated and a total of 450 diabetic patients were identified as FH patients. A noticeable finding of the study is that only one patient (<1%) achieved the LDL-C target recommended for very high-risk patients, but the glycated hemoglobin target level was only reached in 30% of the patients; however, diabetes might had not been newly diagnosed [96]. 

### 10.6. The Role of Obesity in FH Treatment

Obesity is the most common nutritional disorder worldwide and is one of the major risk factors for atherosclerotic CVD [97,98]. It is well known that obesity is intimately associated with dyslipidemia, which is mainly driven by the effects of insulin resistance and pro-inflammatory adipokines. The prominent dyslipidemia in obesity is primarily characterized by increased levels of plasma free fatty acids and TGs, HDL-C level, and abnormal LDL composition [99]. Furthermore, obesity is associated with hypertension and T2DM, which are the major predictors of CVD. All of the above-mentioned abnormalities have been connected with increased CV risk. FH per se is not associated with overweight or obesity, since the pathomechanism of FH does not involve the visceral adipose tissue. It is worthy of note that epicardial fat thickness is significantly increased and related to LDL-C level in patients with FH [100]. However, in the last 3 decades, the worldwide prevalence of obesity has increased 27.5% in adults and 47.1% in children due to complex genetic, socioeconomic, and cultural influences [101]. Consequently, the prevalence of obesity has an increasing significance among FH patients as well. High prevalence of obesity was detected in Hungarian FH patients (42%), both in female (39.4%) and in male (46.1%) [24], similarly to the Romanian prevalence data (36.1%) [44], while it was only 17.1% in a Mexican cohort [47]. It must be noted that glucagon-like peptide-1 receptor (GLP-1R) agonists, approved to treat obesity, elicit robust improvements in overweight, and provide cardioprotection in individuals at risk of or with pre-existing cardiovascular disease [102].

In a study of Mateo-Gallego et al., the effect of weight loss intervention on lipids was different among non-treated overweight adults with FH and familial combined hyperlipidemia, the latter having a better lipid-lowering response to weight loss than FH participants [103]. 

### 10.7. The Effect of Growth Hormone and Its Deficiency on Hypercholesterolemia

Growth hormone deficiency (GHD) is characterized by low levels of growth hormone (GH), which leads to various metabolic disturbances in the lipid and glucose homeostasis and lean and fat mass [104]. Various components of lipid metabolism are altered in GHD, TG, TC and LDL-C levels are elevated, together with decreased HDL-C [105]. Furthermore, GH and/or insulin-like growth factor 1 (IGF-1) are involved in various mechanisms in the regulation of the cardiovascular function; consequently, GHD leads to premature atherosclerosis with increased arterial intima-media thickness and elevates the incidence of cardiovascular morbidity and mortality [106]. 

Childhood-onset GHD is a developmental disorder with characteristic short stature and lifespan-present unfavorable lipid profile; however, the patients’ height and lipid levels usually respond well to therapeutic recombinant human growth hormone (rhGH) [107]. GH decreases lipogenesis and stimulates lipolysis and LDL clearance through LDL-receptor stimulation, which is a direct GH effect as this action becomes absent after IGF-1 treatment [108]. The rhGH therapy reduces LDL-C and TC in GHD children in a dose- and duration-dependent manner. In a few cases in the literature, FH diagnosis was established after inadequate metabolic response to rhGH treatment on lipid profiles with persistently high LDL-C levels [109]. There might be a bidirectional relation between GHD and FH in these children, as the exact effect of hypercholesterolemia on the secretion and function of GH and IGF-1 is not known in detail. 

The lipid profiles of GHD individuals can worsen if they have concomitant FH [109]. GHD leads to dyslipidemia, increased carotid intima-media thickness (cIMT), abdominal obesity, and impaired left ventricular diastolic function. All these factors associated with GHD might accelerate atherosclerosis. The prevalence of GHD is estimated to be between 1:4000 and 1:10,000, while FH has a prevalence of 1:340 (1) [107,110], and both diseases increase cardiovascular risk in childhood. FH and GHD presumably occur rarely together, but FH screening might be beneficial in cases where lipid levels are unusually high or fail to decrease after rhGH initiation in GHD.

### 10.8. The Impact of Hypercholesterolemia on the Reproductive Hormones

The gonadal steroids estradiol and testosterone are fundamental to maintain human reproductive functions and the anabolic state of many tissues. As all steroid hormones are synthesized from cholesterol and optimal steroid production requires an adequate source of cholesterol, a logical concern was that early initiated statins might interfere with steroid hormone production [111]. In vitro studies showed simvastatin to suppress human testicular testosterone biosynthesis [112]. Clinical studies in middle-aged, statin-treated males and females with hypercholesterolemia showed only minimal and clinically irrelevant changes in testosterone and estrogen concentrations [111].

In a study, FH children who were treated with pravastatin for 10 years showed comparable concentrations of testosterone, estradiol, luteinizing hormone, and follicle-stimulating hormone within the normal range compared to their unaffected siblings, the results proving that statins can be safely used even from childhood to prevent premature CVD [111]. 

### 10.9. The Association between Serum Testosterone and Lipids

The association between plasma testosterone and atherosclerotic risk farctors such as obesity, hypertension, diabetes mellitus, and hyperlipidemia is well-known; the European Male Aging Study of 40–79-year-old men reported a higher BMI, waist circumference, systolic pressure, glucose and insulin levels, and lower TC, HDL-C, and LDL-C levels in hypogonadal men as compared to eugonadal men [113,114]. Testosterone replacement therapy has a proved beneficial effect on insulin resistance and lipid metabolism in hypogonadal men with T2DM (the TIMES2 Study) [115]. 

A bidirectional pathogenic link is present between metabolic syndrome (MetS) and hypogonadism, as reduced testosterone predicts MetS and the presence of MetS at study entry increased the risk of hypogonadism development, as described by Popovic et al. Insulin sensitizer treatment in MetS as well as weight loss in obesity resulted in an increase in serum testosterone, while testosterone substitution resulted in the improvement in metabolic derangements characteristic of MetS [116,117].

Cross-sectional studies showed a direct correlation between HDL-C and testosterone, while testosterone treatment also decreased LDL-C levels. Higher Lp(a) was found to be associated with low testosterone in men, Lp(a) improved after treatment with testosterone in men with T2DM in the TIMES2 Study [115,118]. In patients with MetS and late-onset hypogonadism after testosterone substitution, a reduction in cIMT was detected, and the reduction in cIMT appeared to be dose-dependent [119]. 

The importance of the effect of lower testosterone in the general population is based on frequent presentation in the era of “obesity epidemic” as cross-sectional studies have found that 20–64% of obese men have low total testosterone levels [120]. 

### 10.10. Dyslipidemia in Polycystic Ovary Syndrome

Dyslipidemia a common metabolic disorder in women with polycystic ovary syndrome (PCOS). Insulin resistance is fundamental in PCOS development and dyslipidemia in PCOS is present together with insulin resistance. Higher TGs and LDL-C levels together with lower HDL-C concentration in women with PCOS than those of controls was reported in a recent meta-analysis. Furthermore, alterations in LDL quality with an increased proportion of atherogenic small dense LDL or decreased mean LDL particle size were detected. Elevated Lp(a) in PCOS patients has also been consistently reported in diverse studies. The ApoA1 with known cardio-protective effects is significantly lower in women with PCOS, and ApoC1, which increases the postprandial serum lipid level reported to be elevated in PCOS [121,122]. 

According to recent studies, statins can improve lipid abnormalities, insulin sensitivity, inflammation, oxidative stress, and hyperandrogenism in PCOS [123]. Although PCOS is the most common endocrine disorder among women of reproductive age, as depending on the diagnostic criteria used, PCOS prevalence is reported to be between 4% and 19% [124], our knowledge about the prevalence and effect on PCOS among FH women is sparse.

### 10.11. Pregnancy in FH

Women with FH were found to have similar fertility rates compared to women without FH; however, in cases where fertility treatments need to be initiated, a consequent increase in cholesterol and TG levels can develop [125]. 

In pregnant women unaffected by FH, due to the effects of estrogen and progesterone on lipoprotein metabolism, TC normally goes up by 30–40%, together with a rise in LDL-C and HDL-C fraction in the third trimester. The relative increase in TC during pregnancy is the same in healthy women as in FH; however, in the latter case TC easily reaches levels of up to 8–9 mmol/L (309–348 mg/dL) [126].

Statins, ezetimibe and PCSK9-inhibitors are contraindicated when pregnancy is planned, during pregnancy and breastfeeding due to potential teratogenic effects; [127]. in these cases, only non-absorbable bile acid sequestrants and/or LDL apheresis can be considered during pregnancy for the treatment of severe FH [20]. In a recent study, the total length of pregnancy-related off-statin periods before, during, and after the pregnancies of 80 FH women was median 2.3 years with large individual variation [128]. The effect of these treatment interruptions on CV risk in FH women needs to be further elucidated.

In the largest registry study published to date, 2319 births of 1093 women were evaluated with HeFH in Norway. The frequency of preterm delivery (<37 weeks of gestation), low birth weight (<2500 g), and congenital malformations were similar among them compared to the general population of childbearing age. FH women gave birth to more normal-weight infants and fewer high-weight infants compared with women in the general population. The explanation for that phenomenon might be the better nutritional status of FH women as they benefit from regular medical check-ups and lifestyle interventions [125]. 

## 11. Possibilities of Risk Stratification in Familial Hypercholesterolemia

There are various putative clinical and laboratory markers that can provide incremental prognostic information; still, cardiovascular risk stratification of FH patients is an everyday challenge. 

### Imaging

Regarding FH, the advantage of atherosclerosis imaging is to find individuals who may benefit from more treatment beyond therapeutic lifestyle changes and statin therapy. Imaging techniques that have been studied in patients with FH include cIMT assessment by ultrasound, coronary artery calcium (CAC) scoring, and coronary computed tomography angiography (CCTA). Considering cIMT, pediatric FH patients have higher cIMT values compared to age-matched controls who have normal lipid parameters, and as a result, may be a non-invasive marker of CVD risk. Pediatric FH patients on statin medications and followed over 20 years had a slower progression of cIMT and less cardiovascular events compared to their parents [129]. However, cIMT has not been shown to correlate with atherosclerosis in the aorta or coronaries in FH patients [130]. Recently, CAC scoring has been evaluated in FH populations for further CVD risk stratification. Based on the results, CAC was correlated independently and strongly with atherosclerotic CVD events in patients with FH receiving standard lipid-lowering therapy. Therefore, CAC testing might help in stratifying near-term CVD risk in subjects with FH [131]. CCTA has also been shown to help risk stratify asymptomatic patients with FH. Advanced atherosclerotic disease on CCTA correlated to the estimated cardiovascular risk evaluated by the SAFEHEART risk equation. Furthermore, increasing coronary plaque burden as demonstrated in FH patients by CCTA was significantly associated with future coronary events [132].

## 12. Conclusions

The CVD burden from FH differs considerably from patient to patient. Although traditional risk factors contribute significantly to the occurrence of CVD in FH patients, other dyslipidemic conditions might hinder in the shade, and their importance in the atherogenic process might be underestimated (Figure 2). Furthermore, other, still unknown possible genetic and environmental factors can contribute to the development of atherosclerotic disease in these patients. Extended testing for all the above-mentioned traditional and less commonly mentioned risk factors might help in a more precise stratifying of CVD risk in FH patients, and a result, more adequate individual management decision making and population-level resource allocation to help identify patients of high atherosclerotic risk, allowing cost-effective use of novel lipid-lowering agents including PCSK9 inhibitors, to assure a longer and disease-free life for patients with FH.

## Figures and Tables

**Figure 1 genes-13-01158-f001:**
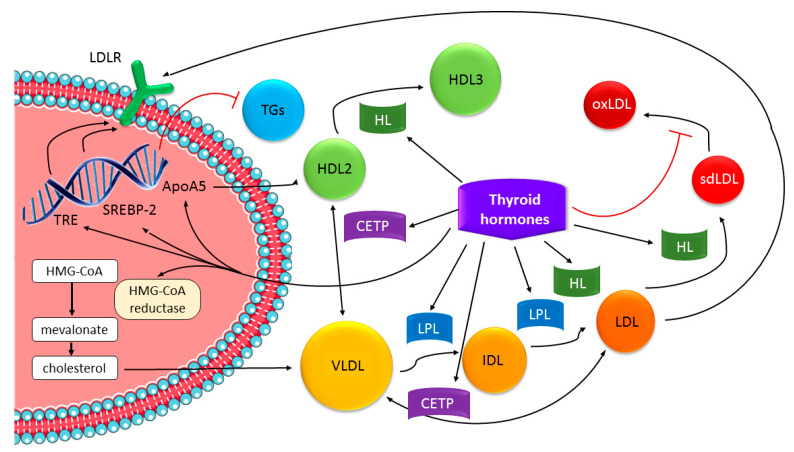
Effects of thyroid hormones on lipid metabolism. Figure modified from [57]. ApoA5, apolipoprotein A5; CETP, cholesteryl ester transfer protein; HDL, high-density lipoprotein; HL, hepatic lipase; HMG-CoA, 3-hydroxy-3-methylglutaryl coenzyme A; IDL, intermediate-density lipoprotein; LDL, low-density lipoprotein; LDLR, LDL receptor; LPL, lipoprotein lipase; oxLDL, oxidized LDL; SREBP-2, sterol regulatory element-binding protein-2;TGs, triglycerides; TRE, thyroid response elements; VLDL, very low-density lipoprotein.

**Figure 2 genes-13-01158-f002:**
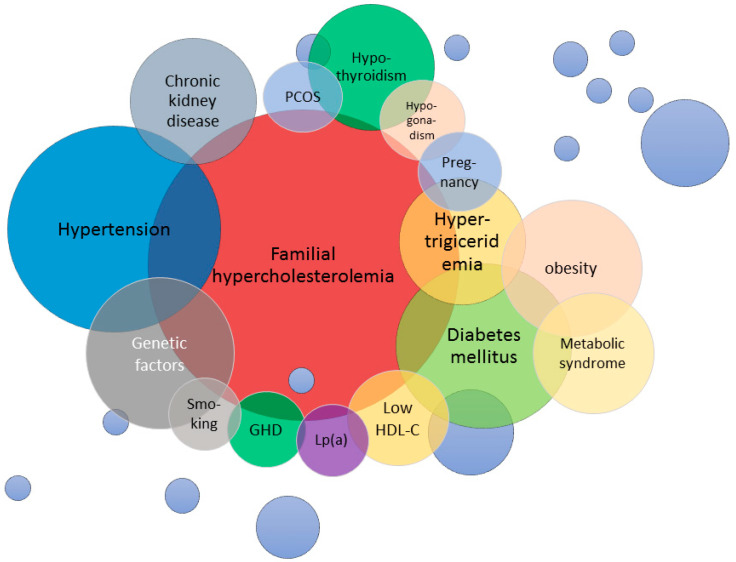
Traditional and non-traditional risk factors for familial hypercholesterolemia.

**Table 1 genes-13-01158-t001:** Selected aspects of clinically relevant endocrine disturbances and their effect on lipid metabolism.

Condition	Clinical Presentation	Underlying Pathophysiology	Clinical Relevance
Hypothyroidism	Elevated triglycerides (TGs) and LDL intermediate-density lipoproteins, apoA and apoB	Increased cholesterol synthesis to degradation rate	After 4–6 weeks of thyroxin replacement therapy corrects dyslipidemia
Diabetic dyslipidemia	Elevated TGs Low HDL-C Small-dense LDL particles	Due to inflammation and increased availability of glucose and/or free fatty acids Increased production of TG-rich lipoproteins	Fundamental compound of elevated cardiovascular risk in diabetes Primary goal: treat to LDL-C to target
Diabetes	Decreased risk for diabetes (especially in patients with LDLR mutations)	Lower intracellular cholesterol levels have a protective effect	Statin therapy does not increase diabetes risk in FH patients Screen FH patients for diabetes with FPG and HgbA1C; weight control
Growth hormone deficiency	Elevated TGs and LDL Low HDL-C	GH decreases lipogenesis, stimulates lipolysis and LDL clearance through LDL-receptor stimulation	rhGH therapy in GHD reduces dyslipidemia and cardiovascular risk
Male hypogonadism	Elevated TGs and LDL Low HDL-C Oxidized LDL	Complex, poorly understood mechanism	Androgen treatment results in a favorable lipid profile
Polycystic ovarian syndrome	Elevated TGs and LDL Low HDL-C Small-dense LDL particles Elevated lipoprotein (a) Decreased ApoA1 Elevated ApoC1	Abdominal obesity, increased lipolysis	Statins improve lipid abnormalities, insulin sensitivity, inflammation, oxidative stress and hyperandrogenism
Obesity	Elevated TGs Low HDL-C Small-dense LDL particles	Postprandial hyperlipidemia, inhibition of hormone sensitive lipase	Lifestyle modification, statins lower TG only marginally, combination with ezetimibe and fibrates to reduce hypertriglyceridemia

ApoA1, apolipoprotein A1; ApoC1, apolipoprotein C1; HDL-C, high density lipoprotein cholesterol; LDL-C, low density lipoprotein cholesterol; LDLR, low density lipoprotein receptor; FH, familial hypercholesterolemia; FPG, fasting plasma glucose; HgbA1C, glycated hemoglobin; GH, growth hormone; GHD, growth hormone deficiency; rhGH, recombinant human growth hormone; TG, triglyceride

## Data Availability

Not applicable.

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
