# Peer review of "Clinical Aspects of Genetic and Non-Genetic Cardiovascular Risk Factors in Familial Hypercholesterolemia"

_genes, 2022, doi:10.3390/genes13071158_

Round 1

Reviewer 1 Report

The authors analysed clinical aspects of genetic and non-genetic cardiovascular risk factors in familial hypercholesterolemia. The review is valuable and interesting.

I have some minor concerns:

1. First of all, the font size is different in several paragraphs (for example the third paragraph in "FH and pregnancy". The article needs formatting.

2. There is a punctuation error in Figure 1 description, a period after the CEPT

3. In my opinion the sentences"In T2DM poor glycemic control results in functional and compositional alterations of small dense HDL, while lifestyle intervention the levels of HDL and large LDL particles increases" and "Among them the elevated plasma LDL cholesterol levels did not affect carbohydrate metabolism, total insulin-mediated glucose disposal, carbohydrate and lipid oxidation rates and the change of free fatty acid concentrations and the change of energy expenditure were similar between the two groups" are very complex and unclear. Could you please reformulate it?

4. In my opinion the review is too long and some parts are difficult to read because of abundant, detailed information. I would suggest to shorten it.

Author Response

Thank you for the review and the valuable comments on our paper. We would like to reply to the reviewer’s comments point by point. Changes in the revised manuscript are marked with track changes.

Comments and Suggestions for Authors

The authors analysed clinical aspects of genetic and non-genetic cardiovascular risk factors in familial hypercholesterolemia. The review is valuable and interesting.

I have some minor concerns:

  1. First of all, the font size is different in several paragraphs (for example the third paragraph in "FH and pregnancy". The article needs formatting.

Response: Thank you for the notification. We corrected the font size throughout the manuscript.

  1. There is a punctuation error in Figure 1 description, a period after the CEPT

Response: Thank you for your comment. We did the correction in the legend of figure 1.

  1. In my opinion the sentences"In T2DM poor glycemic control results in functional and compositional alterations of small dense HDL, while lifestyle intervention the levels of HDL and large LDL particles increases" and "Among them the elevated plasma LDL cholesterol levels did not affect carbohydrate metabolism, total insulin-mediated glucose disposal, carbohydrate and lipid oxidation rates and the change of free fatty acid concentrations and the change of energy expenditure were similar between the two groups" are very complex and unclear. Could you please reformulate it?

Response: According to the reviewer’s suggestion we shortened and rephrased the above mentioned sentences.

  1. In my opinion the review is too long and some parts are difficult to read because of abundant, detailed information. I would suggest to shorten it.

Response: According to the reviewer’s suggestion we shortened and rephrased some parts of the manuscript (marked with track changes).

We thank the reviewer for the highly encouraging and useful comments which have helped us to improve our manuscript.

Reviewer 2 Report

The authors of Clinical aspects of genetic and non-genetic cardiovascular risk factors in familial hypercholesterolemia present an adequate and updated review of aspects of familial hypercholesterolemia and the assessment of cardiovascular risk that must be taken into account in the management of this disease.

In table 2,  (condition) refers to Familial hypercholesterolemia and diabetes, only diabetes should appear in that category.

Author Response

Thank you for the review and the valuable comment on our paper. Changes in the revised manuscript are marked with track changes.

Comments and Suggestions for Authors

The authors of Clinical aspects of genetic and non-genetic cardiovascular risk factors in familial hypercholesterolemia present an adequate and updated review of aspects of familial hypercholesterolemia and the assessment of cardiovascular risk that must be taken into account in the management of this disease.

In table 2,  (condition) refers to Familial hypercholesterolemia and diabetes, only diabetes should appear in that category.

Response: Thank you for the valuable comment. According to the reviewer’s suggestion we corrected the term “familial hypercholesterolemia and diabetes” to “Diabetes” in Table 2.

We thank the reviewer for the useful comment which have helped us to improve our manuscript.